# New Tool Against Tuberculosis: The Potential of the LAMP Lateral Flow Assay in Resource-Limited Settings

**DOI:** 10.3390/cimb47080585

**Published:** 2025-07-23

**Authors:** Marisol Rosas-Diaz, Carmen Palacios-Reyes, Ricardo Godinez-Aguilar, Deyanira Escalante-Bautista, Laura Alfaro Hernández, Ana P. Juarez-Islas, Patricia Segundo-Ibañez, Gabriela Salas-Cuevas, Ángel Olvera-Serrano, Juan Carlos Hernandez-Martinez, Victor Hugo Ramos-Garcia, Esperanza Milagros Garcia Oropesa, Omar Flores-García, Jose Luis Galvez-Romero, Griselda León Burgoa, Manuel Nolasco-Quiroga

**Affiliations:** 1Molecular Biology Laboratory, Multidisciplinary Academic Unit Reynosa-Aztlan Reynosa, Autonomous University of Tamaulipas, Ciudad Victoria 88740, Tamaulipas, Mexico; mar_rodi@yahoo.com.mx (M.R.-D.); jchernandez@docentes.uat.edu.mx (J.C.H.-M.); victor.ramos@uat.edu.mx (V.H.R.-G.); egoropesa@docentes.uat.edu.mx (E.M.G.O.); 2Department of Medical Sciences, Division of Health Sciences, University of Guanajuato, Campus Leon, León de los Aldama 37000, Guanajuato, Mexico; cyapalacios@gmail.com; 3Direction and Division of Research, Hospital Juárez de México, Av. Instituto Politécnico Nacional 5160, Magdalena de las Salinas, Gustavo A. Madero, Mexico City 07760, Mexico; raa_rga@yahoo.com.mx; 4Human Genetics Medical Research Unit, Pediatric Hospital, National Medical Center of the Mexican Institute for Segurity (IMSS), Cuauhtémoc, Mexico City 06720, Mexico; deyusdeyus@gmail.com; 5Laboratory of Cytomics of Childhood, Biomedical Research Center of the East, Puebla Delegation, Mexican Institute of Social Security, Atlixco 74360, Puebla, Mexico; 6Molecular Biology Laboratory of Hospital Clinic Huauchinango, Institute of Social Security and Services for State Workers, Huauchinango 73160, Puebla, Mexico; dra.anapatriciajuarez@gmail.com (A.P.J.-I.); nolasco.quiroga81@gmail.com (P.S.-I.); vivimachorro@hotmail.com (G.S.-C.); angel_serra7@hotmail.com (Á.O.-S.); omar.floresg@issste.gob.mx (O.F.-G.); 7Department of Research of the Regional Hospital ISSSTE of Puebla, Boulevard 14 Sur 4336, Colonia Jardines de San Manuel, Puebla 72570, Mexico; dr.galvez.romero@gmail.com; 8Public Health Laboratory of Puebla, Secretary of Health Mexico, Puebla 772490, Mexico; galvez.buap@gmail.com

**Keywords:** LAMP, tuberculosis, diagnosis, Xpert, AFB smear

## Abstract

Tuberculosis (TB) is a global public health issue requiring early and accurate diagnosis. The loop-mediated isothermal amplification (LAMP) assay is a promising alternative recommended by the WHO for the initial diagnosis of pulmonary TB, particularly in resource-limited settings. This study evaluated the sensitivity and specificity of a commercial LAMP assay for TB detection using 198 samples from different countries including Mexico. The LAMP assay results were compared to the results of standard tests: AFB smear microscopy, cell culture, and Xpert PCR. Across all samples, LAMP showed a sensitivity of 96.20% and a specificity of 84.61%. When compared specifically to “true positives” and “true negatives” (defined by the consistency across the standard tests), LAMP demonstrated 100% sensitivity and 92.30% specificity. For context, the sensitivity of AFB smear microscopy against the culture and Xpert tests was 79.04%. A significant finding was that the LAMP test detected a high percentage (92.5%) of samples found positive by the culture and Xpert tests but negative by the AFB smear, highlighting its ability to identify cases missed by traditional microscopy. This study concluded that the LAMP assay is a sensitive and specific tool for TB diagnosis with potential for rapid and accurate diagnosis, especially in resource-limited areas.

## 1. Introduction

Tuberculosis (TB) is an infectious disease of major public health importance worldwide. Mainly caused by *Mycobacterium tuberculosis*, TB mostly affects the lungs but can spread to other organs. In 2022, the World Health Organization (WHO) estimated that there were about 10 million new cases of TB, resulting in 1.3 million deaths [1,2]. The incidence of TB is disproportionately high in low- and middle-income countries, where factors such as malnutrition and co-infection with human immunodeficiency virus (HIV) and COVID-19 compound its impact [3].

In Mexico, TB is considered a moderate-incidence disease, with an incidence rate of 28 cases per 100,000 inhabitants in 2021 [4]. However, states bordering the United States, such as Baja California, have higher rates due to migrant populations [5]. Baja California, in fact, registers the highest TB morbidity and mortality rates in the country, with an increase in the pulmonary TB morbidity rate from 46.5 in 2017 to 48.9 per 100,000 population in 2021 [3]. In 2023, 2668 TB cases were diagnosed in Baja California, of which 2078 were pulmonary presentations [3]. In 2020, Mexico reported more than 18,000 new cases and almost 2000 deaths from TB. However, it is important to note that underreporting of cases is a prevalent phenomenon in Mexico.

The diagnosis of tuberculosis (TB) is based on a combination of clinical, radiological, laboratory, pathological, and microbiological criteria [6]. The microscopic examination of sputum for acid-fast bacilli (AFB) is the first step, using stains such as Ziehl–Neelsen or Kinyoun [1,7]. Culture tests should also be performed to determine antibiotic susceptibility and genotype. Molecular tests, such as Xpert MTB/RIF and linear probe assays, can identify *M. tuberculosis* DNA and resistance to rifampicin and isoniazid [1,8]. Despite the efficacy of molecular tests, their implementation can be limited by infrastructure and cost. In recent years, the loop-mediated isothermal amplification technique (LAMP) has emerged as a promising tool for TB diagnosis [9].

Currently, there are several modifications of LAMP for visualizing the results: one option is colorimetry using pH changes, and another is a lateral flow assay, which uses nucleic acids labeled with biotin and fluorescein (FITC and FAM, which we use).

The LAMP test has been proposed as a rapid, low-cost alternative that does not require sophisticated equipment and has been recommended by the WHO for the initial diagnosis of pulmonary TB, especially in resource-limited areas [10]. This molecular technique offers a viable alternative to smear microscopy and culture tests, particularly in resource-limited settings, since it is a rapid, inexpensive, and easy-to-perform technique, allowing results to be obtained within 1–2 h [11]. Unlike PCR, LAMP is performed at a constant temperature, which simplifies the process and reduces the need for sophisticated equipment [12].

### Advantages of the LAMP Technique

High sensitivity and specificity: Studies have shown that LAMP has a comparable sensitivity and specificity to the Xpert MTB/RIF assay. The sensitivity of LAMP can reach up to 94.9% in sputum or gastric aspirate samples [13]. In some cases, an even higher sensitivity than microscopy has been reported. A meta-analysis of 26 studies evaluating a total of 9330 sputum samples, including 3069 culture-positive and 6261 culture-negative samples, demonstrated a sensitivity range of 68.7% to 100%, with a mean of 90%, and specificities ranging from 48% to 100%, with a mean of 95.4% [14].Simplicity and speed: LAMP is a simpler and faster process than culture tests or PCR, with results available in 1–2 h [11].Low cost: The cost per test of LAMP is lower than Xpert MTB/RIF, making it more accessible in resource-limited settings [15,16].Inhibitor resistance: LAMP is less susceptible to inhibition by substances present in clinical samples, increasing its reliability [17].Direct visualization: The LAMP reaction produces a significant amount of amplified DNA, allowing the direct visualization of results without the need for complex equipment [11,18].

The WHO recommends the use of LAMP as an initial test for the diagnosis of pulmonary TB in adults and children, particularly in areas with limited resources and a high disease burden [11]. Despite its advantages, it is important to keep in mind that the sensitivity of LAMP may vary depending on the type of sample used and the reference technique. A study conducted in 236 patients (117 suspected TB cases and 119 patients with nontuberculous lung disease) reported a sensitivity of 88.9% and a specificity of 94.4% for the detection of *M. tuberculosis* in adults [10].

The main objective of this study was to evaluate commercial LAMP testing in samples from patients with tuberculosis from different countries without specialized PCR diagnostic equipment.

In this study, we applied LAMP diagnosis to a population in the northern highlands region of Puebla, México, and different countries. We analyzed its sensitivity and specificity, comparing the results with those of AFB smear, culture, and Xpert tests.

## 2. Materials and Methods

Clinical samples were obtained from the tuberculosis diagnostic unit of the Mexican Health Secretariat of the State of Puebla, the ISSSTE Huauchinango Puebla Hospital Clinic, and the World Health Organization (different countries). A total of 198 samples were collected: 3 from gastric juice, 6 from expectoration, and 179 from sputum. Finally, 24 negative controls were obtained, of which 14 were blood samples, and 10 were oral cavity samples. Each sample was divided into three aliquots: one for mycobacterial culture, one for acid-fast bacillus (AFB) smear microscopy, and one for DNA extraction for the LAMP and Xpert test.

The sputum samples were processed using a decontamination and digestion protocol, specifically Petroff’s method, to eliminate the normal bacterial flora and facilitate the growth of mycobacteria. Each sample was treated with a 3% sodium hydroxide (NaOH) solution for 15 min, and then centrifuged and neutralized before inoculation into the culture medium. The processed simples were inoculated into a Löwenstein–Jensen (LJ) culture medium for the primary isolation of *M. tuberculosis* and incubated at 37 °C under aerobic conditions for 15 days. Colony morphology was subsequently examined through microscopic analysis. For confirmation, a sample from a colony was smeared onto a slide, stained using the Ziehl–Neelsen technique, and examined for the presence of acid-fast bacilli.

### 2.1. Bacylloscopy

A smear was prepared by spreading a thin layer of the sputum onto a glass slide, which was then air-dried. The smear was stained with carbol fuchsin, heated to facilitate dye penetration, decolorized with acid alcohol, and counterstained with methylene blue. The stained smear was examined under an optical microscope using a 100× oil immersion objective. Acid-fast bacilli (AFB) appeared as thin red bacilli against a blue background.

### 2.2. DNA Purification from Clinical Samples

The sputum samples were subjected to a decontamination treatment to remove unwanted bacteria and fungi. Samples were treated with a 3% sodium hydroxide (NaOH) solution for 15 min; for purification, a MagAttract HMW DNA Kit (Cat. No./ID: 67563) was used. A total of 200 µL was taken and placed in a 1.5 tube. Then, 300 µL of lysis buffer and 1 µL proteinase k (Promega Cat.# MC5005, Promega, Madison, WI, USA) were added and the sample incubated for 30 min at 60 °C, after which 25 µL of the previously homogenized magnetic beads was added, shaken for 1 min at room temperature, and allowed to stand for 5 min. The solution was then placed in a magnetic rack for 2 min at room temperature. The supernatant was removed and 500 µL of washing buffer was added, perfectly macerated, and allowed to stand for 2 min, after which the supernatant was removed and allowed to dry for 2 min. A total of 50 µL of the elution buffer previously heated to 60 °C was then added, perfectly macerated, and allowed to stand for 2 min. Finally, the supernatant with the purified DNA was collected.

### 2.3. Xpert Test

To determine the presence of *M. tuberculosis* in the samples using PCR, an Xpert MTB/RIF assay (Cepheid Inc., Sunnyvale, CA, USA, REF: GXMTB/RIF-US-10) was used. The procedure was carried out according to the manufacturer’s specifications. Briefly, 1 mL of the sample was taken and mixed with 2 mL of the Xpert sample reagent, vortexed thoroughly, and incubated for 15 min at room temperature. This mixture was transferred into an Xpert cartridge and inserted into the GeneXpert instrument.

### 2.4. Lamp Method

In this study, for the LAMP assay, a commercial TB-DxNet kit from AMUNET (MEXICO) (REF DLTB02) was used. This kit consists of one positive control, one negative control, and Eppendorf tubes containing lyophilized reaction reagents (Bst DNA polymerase, primers, dNTPs, and magnesium), with the primers already included and designed by AMUNET. A tube with diluent reagent and the Bionet multi-visualization system (REF DLTB02), which includes a test strip and its running buffer, are also provided. The assays were carried out according to the manufacturer’s specifications, with the following general steps: 20 µL of diluent solution was added to the tube with the lyophilized reagents, followed by 5 µL of the purified DNA from the sample (or the corresponding control). The mixture was incubated at 65 °C for 30 min in a thermoblock. The visualization of the results was performed using the Bionet multi-system, taking 10 µL of the reaction mixture and adding it to the tube with the running buffer. After briefly shaking, the mixture was applied to the test strip. A positive result is indicated by the appearance of two gray lines: the control line and the specific line of the sequence amplified by LAMP. In negative samples, only the control line will be observed. The absence of the control line or the isolated appearance of the amplification line were considered invalid results.

### 2.5. Statistical Analysis

In this study, a chi-square statistical test was used to analyze the sample types (Table 1). Sensitivity was calculated using the formula Sensitivity = P/(P + FN), where P represents the positive cases correctly detected, and FN represents the false negatives. Similarly, specificity was determined using the formula Specificity = N/(N + FP), where N corresponds to the negative cases correctly identified and FP to the false positives. For the confidence interval, the following formula was used: CI
Sensitivity = Sensitivity ± Z
$\sqrt{Sensitivity(1-Sensitivity})$/No. Samples and C_Specificity_ = Specificity ± Z
$\sqrt{Specificity(1-Specificity})$/No. Samples, where Z is Z-score corresponding to the desired confidence level (1.96 for a 95% Cl). For statistical analysis, we used R-Studio software Version: 2025.05.1.

**Table 1 cimb-47-00585-t001:** Samples collected in Mexico and those provided by the World Health Organization.

Population	Sample	No.	Diabetes	HIV+	*p*-Value ^1^
Mexico	Sputum	77	46	7	*p* = 0.071
	Expectoration	6	4	0	
	Gastric Juice	3	1	1	
	Cepa CMTB	2	0	0	
Negative control	Leukocytes	14	0	0	
Georgia	Sputum	16	0	0	
Moldova	Sputum	32	0	0	
Peru	Sputum	21	0	0	
South Africa	Sputum	24	0	0	
Vietnam	Sputum	18	0	0	

^1^ Chi-square *p*-value. HIV, Human Immunodeficiency Virus. The main comorbidities were diabetes and HIV; no significant difference was observed regarding these comorbidities and origin of the samples.

## 3. Results

A total of 199 patient samples were obtained, distributed as follows: 87 from Mexico and 122 from different countries (Georgia, Moldova, Peru, South Africa, and Vietnam) and 14 negative controls. The sample types were sputum, expectoration, and gastric juice, as shown in Table 1. Derived from the inclusion criteria, all samples obtained from the State of Puebla in Mexico were positive with the AFB smear, culture, and Xpert tests, and comparisons were made with the LAMP test. In the samples received from the World Health Organization, their distribution was as follows: samples positive for AFB smear, culture, and Xpert; samples positive for culture and Xpert; and samples negative for AFB smear, culture, and Xpert. Table 2 shows the following distribution of groups: 127 true positive samples, 40 within the group of other positives (culture and Xpert positives), 24 true negatives (AFB smear, culture, and Xpert negatives), and 20 within the group of other negatives (negative culture and AFB smear). Of the 87 samples obtained from Puebla, Mexico, 51 also displayed type 2 diabetes, which shows a close relationship between diabetes and the risk of contagion with tuberculosis, as described by other authors.

To evaluate the effectiveness of the LAMP assay in detecting tuberculosis, a comparative analysis was performed with the other diagnostic tests. This analysis included all positive controls, both true positives and those classified as “other positives”. Sensitivity was calculated using the formula Sensitivity = P/(P + FN), where P represents the positive cases correctly detected and FN the false negatives. In the study, out of 167 positive samples, the LAMP test yielded three false negatives, resulting in a sensitivity of 96.20%. Similarly, specificity was determined using the formula Specificity = N/(N + FP), where N corresponds to the negative cases correctly identified and FP to the false positives.

The results indicated 44 true negatives and 8 false positives, which translated into a specificity of 84.61%. Additionally, a comparison restricted to the groups of “true positives” and “true negatives” was carried out, obtaining a sensitivity of 100% and a specificity of 92.30% for the LAMP test (Table 3). Finally, when comparing the AFB smear test with the culture and Xpert results, it was observed that the sensitivity of the AFB smear test was 79.04% (Table 3).

## 4. Discussion

In this study, several samples diagnosed with tuberculosis were evaluated using standard tests such as acid-fast bacilli smear (AFB smear), culture, and the commercial PCR test Xpert. Additionally, a commercial LAMP test, DxNet (AMUNET), was used. By using a commercial kit, we avoided manipulation that could lead to reagent contamination and possibly false positives. Furthermore, the samples were handled by healthcare personnel trained in sample handling.

The samples were classified into two main groups: those positive for all three standard tests (AFB smear, culture, and Xpert) and those negative for all three. The results obtained with the LAMP test showed a sensitivity and specificity greater than 90%, as detailed in the corresponding tables. Furthermore, 40 samples were identified as positive for culture and Xpert but negative for AFB smear, representing 23.95% of the total positive samples (167). This finding is consistent with previous studies reporting an approximate sensitivity of 64% when comparing culture and Xpert tests [9].

The LAMP assay proved to be a promising tool for the diagnosis of tuberculosis in the studied population in Puebla, Mexico. The sensitivity and specificity results obtained are comparable to and even higher than those of other diagnostic tests, such as smear microscopy. Overall sensitivity and specificity: When comparing the LAMP assay with the culture and Xpert results (considered the gold standard), a sensitivity of 96.20% and a specificity of 84.61% were obtained. This indicates that the LAMP assay is capable of detecting most positive cases of TB, although it also presents a proportion of false positives. Comparison with true positives and negatives: When analyzing only the true positives (samples positive for AFB smear, culture, and Xpert) and the true negatives (samples negative for AFB smear, culture, and Xpert), the sensitivity of LAMP reached 100% and a specificity 92.30%. This suggests that the LAMP assay has a high performance in detecting confirmed cases of TB and in excluding negative cases. Performance in AFB smear-negative samples: An important finding was that 40 samples were positive for culture and Xpert but negative for AFB smear. This represents 23.95% of all positive samples. The LAMP assay detected a high percentage of these samples (92.5%), suggesting that it may be useful for diagnosing cases of TB that escape detection by smear microscopy. This is relevant since smear microscopy has limited sensitivity, especially in patients with a low bacterial load or in cases of extrapulmonary TB. Comparison with AFB smear: The sensitivity of the AFB smear test with respect to culture and Xpert was 76.04%. This reinforces the idea that smear microscopy has limitations in the detection of TB and that molecular tests like LAMP can improve diagnosis. Additional considerations: It is important to note that the study population included a significant number of patients with type 2 diabetes. Diabetes is a known risk factor for TB, and it could influence the clinical presentation and the results of diagnostic tests. In addition, some samples came from patients from other countries, which could introduce genetic variability in M. tuberculosis strains and affect the sensitivity of the tests. In summary, the LAMP assay proved to be a sensitive and specific tool for the diagnosis of TB in this study. Its ability to detect AFB smear-negative cases and its ease of implementation make it an attractive alternative, especially in resource-limited areas. However, it is important to consider the characteristics of the studied population and conduct further research to validate these results in different contexts.

The LAMP assay demonstrated high sensitivity and specificity for tuberculosis diagnosis, particularly in AFB smear-negative cases and in patients with comorbidities such as type 2 diabetes. Its ease of implementation makes it a viable option in resource-limited settings. However, it presents a proportion of false positives, and its sensitivity may be influenced by the genetic variability of *M. tuberculosis* across different populations. Additionally, this study focused on pulmonary TB and had a limited sample size, requiring further research to assess its performance in extrapulmonary TB and in comparison with other advanced molecular diagnostic tests.

In this study, the majority of samples were sputum; however, it is not limited to this type of samples. One of the limitations of this test is that it does not include the detection of resistant strains. Although most of the reports cited here use colorimetry or fluorescence for visualization, there are other results where they use lateral flow, such as the work of Xinggui Yang et al. 2021 and Yi Wang et al. 2021 [19,20]. In those works, the researchers highlight the ease of handling, and comment that colorimetry is less precise in low DNA concentrations. Although, in the present work, a comparison was not made between these methods, high specificity and sensitivity were obtained as reported in previous works. As shown in Table 4, different studies have been conducted in different countries; however, at the time of writing this paper, no reports of LAMP studies in Mexico were found. In this study, we used samples from international sources and samples from Mexico, and we found no differences in sensitivity and specificity in our results.

## 5. Conclusions

The LAMP assay demonstrated high sensitivity and specificity for tuberculosis diagnosis, particularly in AFB smear-negative cases. Its ease of implementation makes it valuable in resource-limited settings, but false positives and *M. tuberculosis* genetic variability warrant further studies to validate its clinical application.

## Figures and Tables

**Table 2 cimb-47-00585-t002:** The samples were classified as true positives, true negatives, other positives, or other negatives.

Sample	Culture	AFB Smear	Xpert	LAMP
True positives	127 + (100%)	127 + (100%)	127 + (100%)	127 + (100%)
Other positives	40 + (100%)	Negative	40 + (100%)	37 + (92.5%)
True negatives	24 − (100%)	24 − (100%)	24 − (100%)	22 − (91.6%)
Other negatives	20 − (100%)	20 − (100%)	Positive	14 − (70%)

Distribution of samples detected by the LAMP test compared to the other methods. +, Positives. − Negatives. AFB, acid-fast bacilli smear.

**Table 3 cimb-47-00585-t003:** The evaluation of the LAMP test performance using different reference criteria.

Test	LAMP TEST			
	Total	Positive	Negative	DSe/DSp
(a) All Positive (Culture and Xpert)	167	164 (98.2%)	3 (1.79%)	DSe = 96.20 ± 2.8 *****
(b) All Negative (Culture and Xpert)	44	8 (18.2%)	36 (81.81%)	DSp = 84.61 ± 10.6
(c) True Positive (Culture, Xpert, and AFB)	127	127 (100%)	0 (0%)	DSe = 100 ± 0.4
(d) True Negative (Culture, Xpert, and AFB)	24	2 – (100%)	22 (91.60%)	DSp = 92.30 ± 3.4
	AFB TEST			
(e) Positive (Culture and Xpert)	167	127 (76.04%)	40 (23.95%)	DSe = 76.04 ± 6.4

The sensitivity and specificity of the LAMP test vary depending on the methods considered as the gold standard to define true positives and negatives. In general, the LAMP test appears to have high sensitivity, especially when the agreement between the culture, Xpert, and AFB tests is considered a true positive. The specificity is also high, although slightly lower compared to the sensitivity observed in the “true positive” scenario. * Confidence interval. AFB, acid-fast bacilli smear.

**Table 4 cimb-47-00585-t004:** LAMP studies on tuberculosis in recent years.

Country, Year	Detection Method	Title of Study	Sensitivity	Specificity
India, 2025 [19]	Fluorescence	Validation study of a novel, rapid, open-platform, real-time LAMP test for tuberculosis	93.3	94.06
Japan, 2020 [20]	Fluorescence	Diagnostic performance of nucleic acid tests in tuberculous pleuritis	26.5	97.6
India, 2019 [21]	Fluorescence /Lateral flow	Development and evaluation of rapid and specific sdaA LAMP-LFD assay with Xpert MTB/RIF assay for diagnosis of tuberculosis	ND	ND
Thailand, 2019 [22]	Colorimetry	Loop-mediated isothermal amplification for rapid identification of *Mycobacterium tuberculosis* in comparison with immunochromatographic SD Bioline MPT64 Rapid in a high-burden setting	ND	ND
Vietnam, 2018 [23]	Fluorescence	Evaluation of Loopamp™ MTBC detection kit for diagnosis of pulmonary tuberculosis at a peripheral laboratory in a high-burden setting	95.1	80
India, 2017 [24]	Fluorescence Electrophoresis	Evaluation of improved IS6110 LAMP assay for diagnosis of pulmonary and extra pulmonary tuberculosis	97.2	94.4
Morocco, 2016 [25]	Colorimetry	Development and evaluation of an in-house single-step loop-mediated isothermal amplification (SS-LAMP) assay for the detection of *Mycobacterium tuberculosis* complex in sputum samples from Moroccan patients	82.93	99.14
Gambia, 2016 [26]	Fluorescence	Comparison of TB-LAMP, GeneXpert MTB/RIF, and culture for diagnosis of pulmonary tuberculosis in Gambia	99	94
India, 2015 [27]	Fluorescence	Loop-mediated isothermal amplification as an alternative to PCR for the diagnosis of extra-pulmonary tuberculosis	93.3	99.2
China, 2015 [28]	Fluorescence	Real-time fluorescence loop-mediated isothermal amplification (LAMP) for rapid and reliable diagnosis of pulmonary tuberculosis	98	78.3
India, 2013 [29]	Fluorescence	Evaluation of in-house loop-mediated isothermal amplification (LAMP) assay for rapid diagnosis of *M. tuberculosis* in pulmonary specimens	98.4	100
Nepal, 2008 [30]	Fluorescence and turbidimetry	Development of an in-house loop-mediated isothermal amplification (LAMP) assay for the detection of *M. tuberculosis* and evaluation in sputum samples of Nepalese patients	100	94.2

ND, not determined.

## Data Availability

The datasets used in the present study are available from the corresponding author upon reasonable request.

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
