# Peer review of "New Tool Against Tuberculosis: The Potential of the LAMP Lateral Flow Assay in Resource-Limited Settings"

_cimb, 2025, doi:10.3390/cimb47080585_

Round 1
Reviewer 1 Report
Comments and Suggestions for Authors
The manuscript needs some serious editing for clarity and proofreading. Too numerous to list all the errors. For example, M. tuberculosis was not even italicized.
The manuscript needs an introduction of the LAMP method and the commercial kit used. It appears to be a LAMP reaction followed by a lateral flow assay. The authors needs to address the concern of biosafety and contamination in resource-limited settings. If LAMP amplicons are to be transferred to a lateral flow assay, there is a risk of false positives later on. A figure showing the workflow and devices/equipment involved is needed. It would be best if it could show the time required for each step.
The authors should address why a simple colorimetric LAMP was not used. It does not require an additional lateral flow test after amplification. This increases assay time, labor, and cost. The reviewer assumes the work was done mainly because the LAMP/lateral flow assay combination was commercially available.
At line 165. It mentioned Taq polymerase as key ingredient. This enzyme is used in PCR and not LAMP. Primers should not be written as PRIMERS.
It should discuss whether the use of a manual bead-based extraction is suitable in a resource-limited setting. It is rather labor-intensive, although the sputum sample would have been decontaminated first with NaOH.
I do not doubt that LAMP, coupled with a Lateral flow assay, will detect a majority of the TB cases. However, the workflow is mostly manual, and this can lead to user errors or test place contamination. The throughput and cost aspects of the LAMP assay (both reagent and labor) should have been discussed and compared to the Cepheid XPERT method.
Comments on the Quality of English Language
The manuscript needs some serious editing for clarity and proofreading. Too numerous to list all the errors/typos or missing spaces/periods. Letters/numbers were missing in many places, including tables. M. tuberculosis was not even italicized as an example.
Author Response
comments 1: The manuscript needs some serious editing for clarity and proofreading. Too numerous to list all the errors. For example, M. tuberculosis was not The document has been sent for style correction and English editing, and the errors in the tables have been corrected.even italicized.
Response 1 The manuscript was reviewed and edited by a native English speaker and M tuberculosis was italicited in lines 44, 66, 104, 123, 153, 282, 289
comments 2 The manuscript needs an introduction of the LAMP method and the commercial kit used. It appears to be a LAMP reaction followed by a lateral flow assay. The authors needs to address the concern of biosafety and contamination in resource-limited settings. If LAMP amplicons are to be transferred to a lateral flow assay, there is a risk of false positives later on
Response 2
Added LAMP description for lateral flow assay from line 70 to 72, The reason for avoiding contamination from manual handling was described in the manuscript on line 242.
comments 3
The authors should address why a simple colorimetric LAMP was not used. It does not require an additional lateral flow test after amplification. This increases assay time, labor, and cost. The reviewer assumes the work was done mainly because the LAMP/lateral flow assay combination was commercially available.
Response 3
Thank you very much for the observation. Prior to this study, colorimetry systems using pH-based color change were used. However, the data obtained yielded some false positives. Therefore, we decided to use a lateral flow assay, which drastically reduced false positives.
comments 4
At line 165. It mentioned Taq polymerase as key ingredient. This enzyme is used in PCR and not LAMP. Primers should not be written as PRIMERS.
Response 4
The word primers was corrected on line 165.
comments 5
It should discuss whether the use of a manual bead-based extraction is suitable in a resource-limited setting. It is rather labor-intensive, although the sputum sample would have been decontaminated first with NaOH.
Response 5
The kit used in this study has all the reagents included to perform DNA extraction, facilitating use in places with little specialized XPERT equipment. Regarding decontamination with NaOH, it is described in material and methods on line 125.
comments 6
I do not doubt that LAMP, coupled with a Lateral flow assay, will detect a majority of the TB cases. However, the workflow is mostly manual, and this can lead to user errors or test place contamination. The throughput and cost aspects of the LAMP assay (both reagent and labor) should have been discussed and compared to the Cepheid XPERT method.
Response 6
The kit used has all the necessary elements to perform the extraction, amplification by LAMP, and detection by lateral flow, in such a way that the possibility of contamination is reduced.
comments 7
The manuscript needs some serious editing for clarity and proofreading. Too numerous to list all the errors/typos or missing spaces/periods. Letters/numbers were missing in many places, including tables. M. tuberculosis was not even italicized as an example.
Response 7
The document has been sent for style correction and English editing, and the errors in the tables have been corrected.
Reviewer 2 Report
Comments and Suggestions for Authors
Tuberculosis (TB) is a global public health issue requiring early and accurate diagnosis. The loop-mediated isothermal amplification (LAMP) assay is a promising technic, particularly for resource-limited countries, recommended by the WHO for initial diagnosis of the pulmonary TB. This study evaluated the sensitivity and specificity of a commercial LAMP assay for TB detection and demonstrated acceptable sensitivity and specificity.
Unfortunately, there are parts of the paper that require major revision: many typing errors in the Materials and Methods section, unclear lines 116 (CULTURE), 142 (1. 5, 300 µL), and unclear Table 3. Typing errors in the Discussion section and in the Institutional Review Board Statement.
I recommend careful revision of the whole paper.
Author Response
Comments 1: Unfortunately, there are parts of the paper that require major revision: many typing errors in the Materials and Methods section, unclear lines 116 (CULTURE), 142 (1. 5, 300 µL), and unclear Table 3. Typing errors in the Discussion section and in the Institutional Review Board Statement.
Response 1
Thank for you pointing this out,The manuscript underwent a style and English revision. Line 116 the word "culture" was corrected, and line 142 (from µL to µl) was changed.
Reviewer 3 Report
Comments and Suggestions for Authors
LAMP assays have emerged as promising tools for tuberculosis (TB) diagnosis, particularly in resource-limited settings, due to their simplicity, rapidity, and cost-effectiveness.
General comments
Authors should state the limitations, such as sputum samples necessity and drug resistance detection limitations of current LAMP tests.
Tables could benefit from clearer formatting and more descriptive legends.
Optionally, a flowchart summarizing the diagnostic workflow can be helpful.
Minor comments
The study's objectives needs clarification, including the hypothesis and sample description.
The manuscript includes samples from both Mexico and international sources, but it's important to clarify the rationale for including international samples.
Statistical rigor should be included to enhance the robustness of the findings; authors must report confidence intervals for their sensitivity and specificity calculations.
The discussion section acknowledges false positives but doesn't explore potential causes e.g possible cross-reactivity or sample contamination
Mention limitations that include the lack of extrapulmonary TB cases, potential geographic/genetic variability in M. tuberculosis strains, and the absence of cost-effectiveness analysis.
Major comments
Several studies have evaluated the efficacy of TB-LAMP assays for TB diagnosis. Authors need to provide a comparison with different contries' data e.g. Thailand (Gaithuma et al., 2019), Gambia (Rossen et al., 2019), Ghana (Mittal et al., 2024) and India (Mishra et al., 2018)
There are numerous studies with similar pipelines. Authors must do a better discussion comparing their findings with others e.g. Rossen et al., 2019, Promsena et al., 2021, DOLKER et al., 2012, Bojang et al., 2016 and Yadav et al., 2017.
Authors may want to separately present pediatric (paucibacillary) and extrapulmonary cases if present, which are difficult to diagnose currebtly
Comments on the Quality of English Language
Some grammatical errors and awkward phrasing are present, e.g., the word "macerated" is used instead of "vortexed/mixed."
I would recommend professional language editing for clarity and fluency.
Author Response
Comments 1
General comments
Authors should state the limitations, such as sputum samples necessity and drug resistance detection limitations of current LAMP tests.
Tables could benefit from clearer formatting and more descriptive legends.
Optionally, a flowchart summarizing the diagnostic workflow can be helpful.
Response 1
In this study, the majority of samples were sputum; however, it is not limited to this type of sample. One of the limitations of this test is that it does not include the detection of resistant strains.
A better description of the tables was added, in a clearer format, some errors were corrected, and the language edition of the text was revised.
Comments 2
the study's objectives needs clarification, including the hypothesis and sample description.
The manuscript includes samples from both Mexico and international sources, but it's important to clarify the rationale for including international samples.
Statistical rigor should be included to enhance the robustness of the findings; authors must report confidence intervals for their sensitivity and specificity calculations.
The discussion section acknowledges false positives but doesn't explore potential causes e.g possible cross-reactivity or sample contamination
Mention limitations that include the lack of extrapulmonary TB cases, potential geographic/genetic variability in M. tuberculosis strains, and the absence of cost-effectiveness analysis.
Response 2
The objective of the study was described in line 110, the origin of the samples was added in the statistical analysis section in line 197. The calculation of the confidence interval was added to Table 3.
Regarding false positives, it was not evaluated to determine what the possible cause would be. We think that these false positives could be due to an intrinsic factor of the test and not derived from contamination caused by the manipulation of the personnel who perform the test.
n this study, we only included samples from patients diagnosed with pulmonary tuberculosis. We did not evaluate extrapulmonary samples, but it would be interesting to perform this analysis in future research. Regarding genetic variability by region, the test does not seem to show any differences between samples from Mexico and those from other regions of the world. We also added Table 4 on line 325, where specificity and sensitivity results very similar to those we found in this study are observed.
Comments 3
Several studies have evaluated the efficacy of TB-LAMP assays for TB diagnosis. Authors need to provide a comparison with different contries' data e.g. Thailand (Gaithuma et al., 2019), Gambia (Rossen et al., 2019), Ghana (Mittal et al., 2024) and India (Mishra et al., 2018)
There are numerous studies with similar pipelines. Authors must do a better discussion comparing their findings with others e.g. Rossen et al., 2019, Promsena et al., 2021, DOLKER et al., 2012, Bojang et al., 2016 and Yadav et al., 2017.
Response 3
compare our study with what has already been done, we added Table 4 where we placed the most relevant LAMP studies on tuberculosis, adding data such as the origin of the samples and the specificity and sensitivity.
Comments 4
Authors may want to separately present pediatric (paucibacillary) and extrapulmonary cases if present, which are difficult to diagnose currebtly
Response 4
we appreciate all the comments and suggestions you have made. Finally, with regard to using samples from pediatric patients, we did not consider this in this study; however, it could be evaluated in the future, as well as in patients with extrapulmonary tuberculosis.
Round 2
Reviewer 1 Report
Comments and Suggestions for Authors
I appreciate the changes made by the authors. The title should include information that it is an assay of LAMP coupled with a lateral flow assay. The authors stated in their responses that the lateral flow assay portion reduces the risk of false-positive results. Also, I think a few characteristics were not displayed in the statistical analysis section properly (in a pdf file). The font size of the references seem large.
Author Response
Comments 1: I appreciate the changes made by the authors. The title should include information that it is an assay of LAMP coupled with a lateral flow assay. The authors stated in their responses that the lateral flow assay portion reduces the risk of false-positive results. Also, I think a few characteristics were not displayed in the statistical analysis section properly (in a pdf file). The font size of the references seem large.
Response 1: We appreciate your comments and recommendations; we're corrected the statistical analysis and the formulas have been corrected. regarding to sugestión to add "lateral flow assay" to the title, we have already done so, and it Will now read as follows: " New Tool Against Tuberculosis: The Potential of the LAMP Lateral Flow Assay in Resource-Limited Settings" Reviewing the manuscript, we corrected the statistical analysis section, and the formulas are now correct
Reviewer 2 Report
Comments and Suggestions for Authors
All major articles' shortcomings have been addressed. Just one comment concerning Table 2: True positives LAMP should be 127 instead of 1127, and one question: how was 80 % of True negatives calculated for LAMP?
Author Response
Comments 1: All major articles' shortcomings have been addressed. Just one comment concerning Table 2: True positives LAMP should be 127 instead of 1127, and one question: how was 80 % of True negatives calculated for LAMP?
Response 1: Thank You very much for your comment, we have reviewed the manuscript and corrected table 2, changing 1127 to 127. On the other hand for 80%, the calculation for true negatives was made and the correct value is 91.6 %